# Preharvest Management and Postharvest Intervention Strategies to Reduce *Escherichia coli* Contamination in Goat Meat: A Review

**DOI:** 10.3390/ani11102943

**Published:** 2021-10-12

**Authors:** Govind Kannan, Ajit K. Mahapatra, Hema L. Degala

**Affiliations:** Agricultural Research Station, College of Agriculture, Family Sciences and Technology, Fort Valley State University, Fort Valley, GA 31030, USA; mahapatraa@fvsu.edu (A.K.M.); degalah@fvsu.edu (H.L.D.)

**Keywords:** goat meat, food safety, *E. coli*, preharvest management, postharvest intervention

## Abstract

**Simple Summary:**

Goat farms and processing facilities worldwide are primarily small-scale, limited resource operations. Cost-effectiveness and practicality are critical factors to be considered before adopting any pre- and/or post-harvest strategies for pathogen reduction in goat meat. Preharvest management methods in goats that can reduce *Escherichia coli* in meat include minimizing animal stress, selecting diets and feed deprivation times that can reduce fecal shedding of bacteria, and adding tannin-rich feed supplements. In addition, use of appropriate postharvest nonthermal intervention technologies that can reduce microbial loads in carcasses and meat can extend the shelf-life and marketability of goat meat products. Reducing stress prior to slaughter and using nonthermal intervention methods can result in better meat quality and economic returns for producers.

**Abstract:**

Goat meat is the main source of animal protein in developing countries, particularly in Asia and Africa. Goat meat consumption has also increased in the US in the recent years due to the growing ethnic population. The digestive tract of goat is a natural habitat for *Escherichia coli* organisms. While researchers have long focused on postharvest intervention strategies to control *E. coli* outbreaks, recent works have also included preharvest methodologies. In goats, these include minimizing animal stress, manipulating diet a few weeks prior to processing, feeding diets high in tannins, controlling feed deprivation times while preparing for processing, and spray washing goats prior to slaughter. Postharvest intervention methods studied in small ruminant meats have included spray washing using water, organic acids, ozonated water, and electrolyzed water, and the use of ultraviolet (UV) light, pulsed UV-light, sonication, low-voltage electricity, organic oils, and hurdle technologies. These intervention methods show a strong antimicrobial activity and are considered environmentally friendly. However, cost-effectiveness, ease of application, and possible negative effects on meat quality characteristics must be carefully considered before adopting any intervention strategy for a given meat processing operation. As discussed in this review paper, novel pre- and post-harvest intervention methods show significant potential for future applications in goat farms and processing plants.

## 1. Introduction

Enterohemorrhagic *Escherichia coli* (*E. coli*) is considered one of the most economically important food-borne pathogens. Much of the research has focused on post-slaughter sanitation to improve the safety of meat products, and as a result, various strategies are practiced in meat plants to reduce carcass contamination. In recent years, researchers have also been working on developing intervention strategies in the live animal prior to slaughter to reduce foodborne pathogens. Because fecal shedding is correlated with carcass contamination, reducing bacterial loads in the gastrointestinal tracts of live animals is important in the production of safe and wholesome food products. 

It is widely recognized that hygienic risks at slaughterhouses should be assessed in reference to the number of organisms indicative of fecal contamination [1]. The Agricultural Marketing Service of the US Department of Agriculture has established 500 CFU g^−1^ as a critical limit for generic *E. coli* in red meat such as beef [2]. Although *E. coli* organisms are normal inhabitants of the gastrointestinal tracts of ruminants, some pathogenic strains can cause hemorrhagic colitis in humans [3]. *Escherichia coli* O157:H7 is the most common enterohemorrhagic *E. coli* serotype implicated in many outbreaks of bloody diarrhea and the hemolytic–uremic syndrome resulting in kidney failure. 

Goat meat is one of the most widely consumed meats in the world, especially in Asia and Africa, and importation of goat meat into the US has steadily increased mainly due to increased demand by ethnic consumers. However, research on intervention strategies to reduce pathogens in goat carcasses, cuts, and products is very limited. Data available on preslaughter intervention strategies to reduce foodborne pathogens in goats are scanty. 

Dietary manipulation, feed deprivation duration prior to slaughter, supplementation with high tannin-containing diets, minimizing animal stress, and live animal washing have been studied as possible pre-harvest intervention strategies to reduce *E. coli* in goats. Research on postharvest methods in goats have included treating goat meat with ozonated water, electrolyzed oxidizing water, ultraviolet light, sonication, organic acids, and organic oils, to name a few. In addition, nonthermal hurdle technologies with different treatment-time combinations have also been found to be promising. The body of literature currently available on the various low-cost pre- and post-harvest intervention strategies to reduce *E. coli* populations in goats was reviewed in this work to benefit smallholder farmers and small-scale meat processors and retailers worldwide. 

## 2. Prevalence of *E. coli* in Goats

Several countries from different continents have reported *E. coli* O157:H7 in humans. Although this organism has been isolated from several animal species, ruminant livestock species are regarded as natural carriers of *E. coli.* These food animals typically do not show any clinical signs while shedding *E. coli* through feces. Isolation of *E. coli* O157:H7 from goats was first reported in 1994 from a human outbreak of *E. coli* O157:H7 in United Kingdom [4]. 

Researchers have reported different prevalence rates in different countries. According to Dulo et al. [5], *E. coli* O157:H7 was present in 2.2% of cecal content samples and 3.2% of carcass swab samples obtained from goats in Somali region of Ethiopia. The researchers suggested that poor hygiene and slaughter practices may cause contamination of meat and human health risks as consumption of raw meat is a common practice in Ethiopia. A comparison of *E. coli* O157:H7 contamination rate among beef, lamb, and chevon samples from slaughter plants and retail outlets in Addis Ababa, Ethiopia, showed that beef was the most frequently contaminated meat, followed by sheep and goat meat [6]. This study also revealed that contamination rates were higher at retail shops than at slaughterhouses for beef (21.9 vs. 4.7%), sheep (10.9 vs. 6.3%), and goat (9.4 vs. 6.3%) carcass and meat samples. Sixty percent of goat meat samples collected from different markets in Dhaka, Bangladesh has been reported to be positive for *E. coli* O157:H7 strains, although no official infections have been reported either due to improper tracking of outbreaks and causative organisms or to possible acquired immunity in the population [7]. Another report from Bangladesh also showed that prevalence of multidrug-resistant *E. coli* was higher in younger goats than older goats. The authors also reported that the prevalence of drug-resistant *E. coli* was higher in goats raised in poor hygienic conditions than those raised in good hygienic conditions, and higher in goats with recent history of transportation [8]. Shiga toxin-producing *E. coli* prevalence in Jordan was found to be greater in intensively reared goats with occasional grazing (65%) than in extensively reared goats with year-round grazing (50%) [9]. 

Preharvest meat goat management, slaughter and processing, and postharvest carcass and meat handling methods vary among different countries and regions. The *E. coli* counts on goat skin prior to slaughter generally range from 2.2 to 2.5 log_10_ CFU cm^−2,^ and those on carcasses before washing range from 2.1 to 2.3 log10 CFU cm^−2^ [10,11]. It is not clear to what extent factors such as sex, age, and breed can influence *E. coli* shedding in goats. A generic flow diagram of various steps involved in pre- and post-harvest phases of chevon production is included to illustrate potential intervention points (Figure 1). 

## 3. Pre-Harvest Intervention

Studies involving preharvest intervention to reduce *E. coli* population in goats are very limited (Table 1). Before recommending any management method as a viable means of controlling potential transfer of organisms from the gastrointestinal tract to the skin, carcass, and meat, it is important to consider how an intervention strategy can affect animal welfare, productivity, and product quality aspects [35]. Since most goat meat producers around the world are smallholder farmers, it is also important to factor in the economic implications for farmers. 

### 3.1. Meat Goat Management and Productivity

Small ruminant management practices that increase animal stress may alter the normal course of conversion of muscle to meat, leading to inferior meat quality [39]. Chronic stress in animals can negatively affect animal growth, weight gain, and immune function, which in turn can lead to poorer health, carcass yield, and economic returns [40]. Animal management during the 24-h period prior to slaughter is crucial not only for animal welfare reasons, but also for profitability. Poor preslaughter handling practices in goats can increase dehydration and live weight shrinkage and decrease carcass yield and meat quality [21], in addition to increasing the chances of fecal contamination of skin and carcasses [41]. Preparation of goats for slaughter generally involves loading onto a trailer, transportation, unloading, feed deprivation, exposure to novel environments, noise and vibration, disruption of social groups, and changes in temperature and humidity, all of which will have to be carefully considered and managed appropriately to minimize negative effects [40,42]. 

Some researchers who evaluated preharvest intervention methods to reduce fecal contamination have also evaluated other aspects of economic importance. For example, Kannan et al. [10] reported that skin bacterial counts can be significantly reduced in goats by preslaughter spray washing without increasing animal stress. 

### 3.2. Dietary Regimens and E. coli Populations

A considerable amount of data is available on the effect of diet on fecal shedding of generic *E. coli* in cattle and sheep; however, studies on goats are very limited. Finishing beef cattle and lambs in the US are often fed grain rations to improve productivity. Meat goats are raised primarily on pastures with grain supplements in some cases. Dietary starch is protected from ruminal microbial degradation by a protein called zein [3]. Because there is less pancreatic amylase activity in the small intestines of ruminants, much of the dietary starch reaches the cecum and colon, where it undergoes a secondary microbial fermentation [43]. Fermentation of starch by bacteria, including *E. coli,* in the cecum and colon produce volatile fatty acids that could reduce the pH of the colonic digesta and inhibit *E. coli* [3]. According to Gutta et al. [13], the pH values of the rumen contents of hay-fed small ruminant animals were higher (7.08) than concentrate-fed animals (6.43). A similar effect was also noticed in colon contents, with pH values of 7.02 and 6.56 in hay-fed and concentrate-fed animals, respectively. Regardless of these unfavorable conditions, research shows *E. coli* grows in the intestinal tract of cattle fed high-grain rations. A suggested explanation for higher *E. coli* counts despite lower pH is the fermentation of the easily assimilable carbohydrate portion of concentrate diets in the rumen, so that substances that facilitate microbial growth likely become easily available. Moreover, Gutta et al. [13] observed that in goats and sheep, *E. coli*, coliform, *Enterobacteriaceae*, and total plate counts were more associated with pH in the colon than in the rumen. The authors observed a negative correlation tendency between *E. coli* numbers and colon pH. A similar effect was reported by Scott et al. [44] in cattle, which indicates that fermentation of starch in the colon produces nutrients necessary for both bacterial growth and pH decline. Acid resistance in *E. coli* organisms in the colon is greater when animals are fed a concentrate diet rather than a hay diet [45]. Russell et al. [46] speculated that by modifying the concentration of undissociated volatile fatty acids (VFA), colonic pH may play a role in regulating the resistance of *E. coli* to low pH values. It is not clear if the negative correlation tendency between *E. coli* counts and colon pH observed by Gutta et al. [13] was due to the acid resistance of organisms in the colon. The site of persistence of *E. coli* in adult ruminant animals appears to be the colon [47]. 

Shedding of *E. coli* by ruminants is influenced by the diet [48]. There are conflicting reports on the effects of forage and concentrate diets on fecal shedding and colonization by *E. coli* in the gastrointestinal tract of ruminants. Gutta et al. [13] reported that concentrate-fed goats and sheep had higher *E. coli* (6.44 vs. 4.01 ± 0.468 log_10_ CFU g^−1^), total coliform (6.74 vs. 4.16 ± 0.469 log_10_ CFU g^−1^), *Enterobacteriaceae* (6.93 vs. 3.83 ± 0.651 log_10_ CFU g^−1^), and total plate counts (7.79 vs. 7.28 ± 0.170 log_10_ CFU g^−1^) in the rectum than the hay-fed animals. The authors suggested that microbial loads in the gastrointestinal tract of goats and sheep can be reduced by feeding hay for four days before slaughter, although diet did not have any effect on skin bacterial contamination in goats and sheep [11]. To determine the effect of diet on gastrointestinal tract bacterial populations in goats, Lee et al. [37] conducted a feeding trial involving three dietary treatments: 90-day alfalfa hay alone, 90-day concentrate alone (18% crude protein), or 45-day alfalfa hay diet followed by 45-day concentrate. The authors found that diet did not have any effect on rumen *E. coli* population; however, diet had a significant effect on rectal *E. coli* counts with the population being lower in hay-fed goats compared with concentrate-fed goats. 

A change in ruminant diet from concentrate to hay in the days before slaughter has been shown to reduce fecal shedding of bacteria, although this effect was also not consistent among studies. Shifting cattle from a high grain (90% corn/soybean) diet to 100% timothy hay resulted in a significant reduction of generic *E. coli* in feces [45]. Gregory et al. [14] reported that switching cattle from pasture to hay 48 h prior to slaughter significantly reduced the *E. coli* burden throughout the gut. The authors further speculated that the increased intestinal *Enterococci* populations due to hay feeding can inhibit *E. coli* populations. However, a diet change from alfalfa pellet to poor quality forage has been shown to increase *E. coli* O157:H7 fecal shedding in experimentally infected sheep [49]. Sheep shifted from a 50:50 corn/alfalfa ration to low quality hay shed greater populations of *E. coli* O157:H7 than animals fed exclusively on the corn/alfalfa ration [48]. These discrepancies in results from different studies on diet switching from concentrate to hay reveal that other factors such as breed, age, season, feed quality, and consumption could also be involved in determining *E. coli* shedding in ruminants. 

### 3.3. Feed Withdrawal and E. coli Populations 

It is a common practice to withhold feed in meat animals during periods of preslaughter transportation and holding, to minimize the difficulties of handling overfilled guts during evisceration and incidences of gastrointestinal tract rupture and soiling of hide/skin. Since the primary sources of carcass contamination with enterogenic pathogens are the hide and gastrointestinal tract, feed withdrawal may be an important step in reducing the bacterial load of feces excreted during the preslaughter period. 

Conflicting reports have been published on the effects of feed deprivation on fecal shedding of *E. coli.* Gutta et al. [13] reported that feed deprivation for 24 h increased *E. coli*, total coliform counts, and *Enterobacteriaceae* counts in the rumen of Kiko × Spanish goats and Dorset × Suffolk sheep, compared with 12 h deprivation and with no significant change in the pH of rumen liquor. However, no difference in *E. coli* or total coliform counts were observed on sheep and goat (Kiko × Spanish) carcasses as a result of feed deprivation time [11]. In contrast, experimentally infected adult sheep showed no increase in fecal shedding of *E. coli* O157:H7 during feed withdrawal [48]. According to Vanguru et al. [36], Boer × Spanish goats subjected to either 0, 9, 18, or 27 h of feed deprivation prior to slaughter showed that the 27-h feed deprivation group had higher rumen pH (6.95) than those at 0 h (6.23) or 9 h (6.46) feed deprivation, although there were no differences in the microbial counts of rumen or fecal samples among the groups. The authors concluded that feed deprivation time alone up to 27 h may not significantly influence gut, skin, or carcass microbial loads. 

Low pH is not favorable for growth of naturally occurring *E. coli*; however, the organisms can survive low pH and start growing again if the pH becomes favorable [46,50]. The effect of fasting on *E. coli* growth in the ruminant digestive tract is mediated by a reduction of VFA concentration and the subsequent increase in the pH of the digestive tract contents [51]. *Escherichia coli* grew best in rumen liquor in vitro when the concentration of VFA was less than 25 mM and the pH was 7.2, and the growth of *E. coli* was completely arrested when VFA concentration was greater than 75 mM. Also, a linear decrease in *E. coli* growth as pH declined, and zero growth at a pH 6.0, were observed [51].

The inconsistencies in results from different studies on the effects of diet and feed deprivation on gut *E. coli* populations and fecal shedding could be due to various other factors such as environment, season, breed, age, previous feeding regime, and bacterial strain [41]. Bacterial shedding by ruminants is very complex and can be affected by various interrelated factors, such as extent of attachment to intestinal mucosa, ease of detachment from mucosa, distribution and growth in solid-liquid phases of gut contents, and passage rates through different parts of the gastrointestinal tract [52,53]. The type of feed can affect the consistency of gastrointestinal tract contents and passage rate. Grazing on young pasture often results in diarrhea in ruminants, and the abbreviated stay of fecal contents in the colon also prevents reabsorption of water, which could result in purging and potentially dirty hindquarters in ruminants [21].

### 3.4. Tannin-Containing Feed Supplements

A preharvest diet containing brown seaweed (*Ascophyllum nodosum*) extract has been reported to reduce fecal shedding of *E. coli* and other enteric bacteria. Supplementation of brown seaweed extract in a conventional grain diet reduced *E. coli* counts both in fecal samples and hide swipes in feedlot Angus steers [15]. Enterohemorrhagic *E. coli* and *Salmonella spp.* populations were significantly reduced by brown seaweed extract supplementation in feedlot steers [54]. A brown seaweed extract-containing diet has been shown to result in enhanced antioxidant status and immune function in live animals, and improved meat quality characteristics and food safety [16,55,56,57]. 

Among the several biologically active compounds present in *A. nodosum*, phlorotannins, a group of polyphenols, have been shown to possess marked antibacterial activity. The mode of action has not been understood fully yet, although there is evidence that the phlorotannins could be at least partially responsible for its antibacterial property [58]. The authors reported that phlorotannins isolated from *A. nodosum* had significant antimicrobial activity that affected several rumen bacteria. The antibacterial effects of terrestrial tannins are attributed to several mechanisms, including inhibition of oxidative phosphorylation and extracellular microbial enzymes, dysfunction of cell membranes, and deprivation of substrate metal ions and minerals [59]. Since the mechanism involved in the antibacterial activity of phlorotannins is not fully elucidated, Wang et al. [58] speculated that the antimicrobial mechanisms could be similar in both phlorotannins and terrestrial tannins. In addition, the authors demonstrated, using transmission electron microscopy, that phlorotannins can inhibit bacterial growth by interfering in the cell membrane functions. There is also evidence that phlorotannins possess more potent antibacterial activities compared with hydrolysable terrestrial tannins or condensed tannins, probably due to the greater number of hydroxyl groups [58] and the degree of phloroglucinol polymerization [60] in phlorotannins. 

Lee et al. [38] fed Kiko × Spanish goats either ground sericea lespedeza (SL; *Lespedeza cuneata* (Dum-Cours) G. Don) or bermudagrass (BG) hay at 75% of daily intake with a corn-based supplement (25%) for 14 weeks before slaughter to see the effect on fecal *E. coli* shedding. They reported that high dietary condensed tannins in SL increased *E. coli* and total coliform populations in the rumen without affecting the populations in the rectum. The sericea hay-fed goats had higher rumen pH due to lower production of total VFA, a condition that favored *E. coli* and total coliform growth in the rumen, although such an effect was absent in the rectum. However, the authors found that the total plate count in the rectum samples from goats fed sericea was lower than that from goats fed bermudagrass hay. In another study, Mechineni et al. [12] fed Spanish goats either BG, SL, or a combined SL + BG diet for eight weeks. Half of the goats from each paddock were subjected to a 3 h transportation stress. The results indicated that dietary treatment did not affect gastrointestinal tract, skin, and carcass microbial populations or meat quality. Transportation stress also had no significant effect on gastrointestinal tract, skin, and carcass microbial populations or on meat quality.

### 3.5. Essential Oil-Containing Feed Supplements

Dietary supplementation of essential oils derived from plants such as thyme, rosemary, and sage have been evaluated in ruminants for their antimicrobial and rumen modifying activities. These compounds are secondary metabolites that often contain terpenoids and phenylpropanoids [61]. 

The antibacterial effects of essential oils are primarily due to their hydrophobic properties, which enable them to disrupt the bacterial cell wall and mitochondria by partitioning lipids. This results in changes in membrane integrity, ion transport processes, and cell osmotic pressure [62]. Although essential oils are more effective against gram-positive bacteria, low molecular weight molecules of essential oils can get through the cell wall of gram-negative bacteria via diffusion through membrane proteins, thus disrupting the membrane integrity [63]. The bacterial cells can offset these effects using ionic pumps; however, the high energy cost involved results in the slowing down of ATP synthesis and microbial growth, and the eventual death of bacteria cells [64]. In addition to bacterial cell membrane disruption, essential oils have also been reported to coagulate cell contents by protein denaturation [65], interact with proteins through hydrophobic interactions [66] and through other functional groups [66,67], and inhibit enzymatic activity [68]. 

The antimicrobial properties of the herb thyme (*Thymus vulgaris*) are well researched. In vitro studies have shown that the essential oils of thyme inhibit *E. coli* growth [17] and modify rumen VFA concentrations [69]. The primary constituents of thyme essential oils are thymol and carvacrol [70]. Thyme also contains caffeic acid, which has antibacterial properties [71]. Rosemary and thyme essential oils can inhibit certain rumen bacteria involved in biohydrogenation due to a high polyphenol content, resulting in an increase in polyunsaturated fatty acids in rumen digesta [18]. However, the results on the effects of these compounds on ruminal fermentation have not been consistent.

### 3.6. Spray Washing

Small ruminants with dirty skin or hair/wool presented for slaughter can pose food safety risks, since the degree of visible contamination on the hide or skin has been shown to affect the contamination levels of the resultant carcass. The skin of a live animal becomes contaminated with both pathogenic and nonpathogenic microorganisms derived from a wide range of sources such as feces, soil, water, and vegetation [72]. Animals can spread the contamination to other cleaner animals during transport and holding, either directly via body contact, indirectly via contact with contaminated floors, or both [73]. The presence of dirt or feces on animals can potentially lead to the transfer of microbes to the carcass and meat and to the slaughter equipment. 

Methods to deal with excessively dirty animals during the preslaughter period vary in different countries, and these may include isolation of these animals as unfit for slaughter in their present condition, provision of a clean-up period by allowing to graze pasture, or inclusion of a swim-washing or spray-washing step prior to slaughter [21]. However, spray washing live animals prior to slaughter is not permitted in some countries, although studies have indicated this treatment reduces hide/skin fecal contamination and microbial loads [10,22]. 

Kannan et al. [10] found that the aerobic plate counts on the skin were the same for both spray-washed (1 min with potable water) and unwashed Spanish goats, but the counts were significantly less in the treated group when sampled after washing. However, skin *E. coli* counts did not decrease significantly due to spray-washing treatment. Spray washing also did not influence carcass *E. coli* or aerobic plate counts. Several factors can influence the antimicrobial effect of spray-washing, including duration of the spray-washing treatment, pressure of the spray nozzle, and whether or not the water contains any antibacterial agents. Byrne et al. [22] observed that a 3-min spray-washing treatment in cattle yielded better results compared with a 1-min washing treatment in reducing hide *E. coli* populations. It is likely longer durations and increased water pressure during spray- washing can increase discomfort and stress in live goats. A study by Kannan et al. [16] showed that applying a two-step spray-washing treatment in Boer × Spanish goats after exsanguination that comprised of 1 min washing with potable water (to remove fecal material) followed by 1 min with chlorinated water (to kill bacteria) decreased both skin *E. coli* and aerobic plate counts. In practical situations, this strategy would allow increasing of the duration and pressure of washing, if needed, since the washing treatment is imposed after bleeding.

### 3.7. Animal Behavior and Physiology 

Minimizing stress in goats during loading, transportation, unloading and holding prior to slaughter is very important in reducing meat hygiene risks. It is important to consider goat behavior and physiology in response to a preharvest intervention method before recommending it as a viable pathogen reduction technology. Defecation is a frequently noticed stress response in cattle. Stress combined with light physical exercise can accelerate gastrointestinal tract emptying, while vigorous exercise slows gastric emptying [21]. Ruminants infected with pathogenic bacteria can harbor the organisms in the cecum, where they multiply. Stress can speed up emptying of the cecum into the colon and increase the rate of excretion, which can in turn increase the likelihood of contamination of the hide/skin of pathogen-free animals and increasing the food safety risk in the slaughter plant [21,74]. 

Several studies have evaluated stress levels in goats in response to preslaughter management practices such as transportation, feed deprivation, social isolation, and spray washing. Loading meat goats onto a transport trailer increases stress as indicated by increasing plasma cortisol and glucose concentrations. A 2½-hour transportation combined with an 18-h feed deprivation increases plasma cortisol concentrations in goats [40]. However, when goats are feed-deprived without transportation, cortisol concentrations do not increase [42]. While feed deprivation can facilitate evisceration without rupturing the digestive tract during the slaughter process, it can also contribute to live weight and carcass shrinkage losses and result in economic losses to the producer. 

Social isolation of goats when they are moved through the single file race with individual compartments just prior to slaughter can increase stress. Isolation of goats from their social group for 15 min increased plasma cortisol concentrations; however, stress levels did not increase in socially isolated goats that could maintain visual contact with other goats [42]. Any intervention method used to treat excessively dirty animals prior to slaughter such as spray washing is likely to involve social isolation and further increase stress levels in goats. However, spray washing alone does not significantly increase stress in goats compared with unwashed controls if the water pressure used is carefully controlled to prevent any visible discomfort in the animals [10]. 

An important factor that can influence the efficacy of spray washing in small ruminants is the length of hair/wool and smoothness of the animal’s coat. Kannan et al. [11] observed that under identical preslaughter management conditions, sheep may be more prone to skin bacterial contamination than goats. This is probably because the goat breeds used for meat production in the US generally have smoother and thinner coats. In addition, the season of the year and the behavior of the animals in the holding pens can influence the extent of skin contamination and the efficacy of spray-washing treatment. Sheep tend to spend more time lying down on the concrete floor of the holding pens compared to goats in winter, probably because goats with thin smooth coats cannot tolerate cold concrete floors for extended periods [11]. Since animals held in holding pens continue to defecate, they can pick up fecal materials from the contaminated floor when they lie down, the extent of which can be greater when animals have rough long coats [11].

## 4. Postharvest Intervention

Post-harvest intervention can be at any point in the processing line of carcasses after dressing is completed or in the product movement path before they reach consumers (Figure 1). This can involve applying intervention technology to whole carcasses, primal cuts, retail cuts, or ready-to-cook cuts such as bone-in cubed or boneless cubed meat. Goat meat is usually marketed as fresh meat, and marketing further-processed goat meat is extremely rare in the US. Postharvest intervention techniques can be broadly classified into thermal and nonthermal methods. Thermal techniques are effective in reducing pathogen counts on meat; however, they could adversely affect the quality characteristics of fresh meat due to heat-induced changes in muscle structure, composition, and biochemistry. Researchers have focused on nonthermal novel intervention methods that are effective in reducing pathogen counts (Table 2) without having any noticeable negative effects on fresh meat quality due to temperature increase [30]. It is essential to evaluate the effects of nonthermal technology used on fresh meat properties, such as appearance, color, odor, and lipid oxidation, and on cooked meat properties, such as texture, juiciness, and flavor. In addition, some low-cost technologies can leave chemical residues in meat, in addition to negatively affecting the flavor of meat. Technologies that cause significant negative effects on fresh meat quality that are easily discernable by consumers cannot be considered as viable intervention methods, even if they are efficient in reducing microbial numbers. Several studies assessed the effectiveness of these nonthermal technologies on poultry, red meat, and sea foods. The results of these studies show that these novel nonthermal technologies can be potentially applied to any meat, regardless of animal species, and therefore could be easily adopted in small-scale goat meat processing units. In the following sections, those methods that qualify as nonthermal, low-cost technologies with practical applications in goat meat processing are reviewed.

### 4.1. Organic Acids

Spray treatment of food animal carcasses using various decontaminant solutions has been extensively studied; however, studies in chevon carcasses have been limited. Goats have very limited subcutaneous fat deposition and more visceral fat accumulation. As a result, a dressed goat carcass is typically devoid of fat coverage compared to the extensive subcutaneous fat seen in beef or lamb carcass. Therefore, spray-washing parameters specific to goat carcasses may be required for practical applications. Spray-washing treatments studied to decontaminate food animal carcasses have included ozonated water, chlorinated water, trisodium phosphate, purified water, hot water, and organic and other acid solutions, to name a few. Adding chlorine to water used for spray washing appears to have little added advantage over water alone, based on research results [25]. Organic acids (lactic, acetic, and citric) were found to be very effective for broad-spectrum decontamination of meat carcasses and were proven to be better than many other compounds studied. An advantage of using organic acids over other intervention strategies is that residual antimicrobial activity is seen over extended storage time, although carcass decontamination may not necessarily improve the safety of meat cuts [26]. Among the organic acids, lactic acid is preferred due to its nonirritant characteristic, in addition to its decontamination properties. 

The decontamination efficacy of organic acid spray washing depends on various factors, including strength of acid, solution temperature and contact time, spray pressure, and acid adaptation of organisms [27,28]. A 1.5 to 2.5% solution of food grade organic acid is recommended as ideal for carcass spraying. Microbial population reductions of 1 to 4 log CFU cm^−2^ have been reported with the use of varying concentrations [77]. High spray pressure has the potential of damaging the carcass surface and allowing the organisms to penetrate into the carcass, which could contaminate the meat [78]. Higher water temperature can negatively affect carcass surface color, although Dorsa [25] reported that a water temperature range of 70–96 °C did not permanently affect carcass appearance. Studies in goat and sheep carcasses have shown that a spray-wash treatment with a 2% lactic acid and 1.5% acetic + 1.5% propionic acid combination can reduce total viable counts from 0.52 to 1.16 log units with minimal changes in meat color and odor scores [79], and can extend shelf life to 8–11 days, compared to a shelf life of 3 days in untreated samples. However, lactic acid and acetic acid treatment application directly on meat cuts may result in permanent adverse changes in sensory properties [26].

### 4.2. Ozonated Water

Ozone is an effective antimicrobial for the treatment of meat due to several advantages, including its reactivity, penetrability, and spontaneous decomposition into a nontoxic product. Several researchers have assessed the antimicrobial efficacy of ozone, as it decomposes to oxygen continuously without leaving any residue in meat [29,31]. After production, ozone water must be used within 15 min for the decontamination of foods [80] as its half-life at room temperature is short due to instability in aqueous solutions [81]. The antimicrobial activity of ozonated water depends on temperature and pH, and on the presence of dissolved compounds such as sugar, minerals, surfactants, and organic matter [82]. Although several studies reported use of ozone in gaseous or aqueous form on fruits, vegetables, and greens, studies in meat, especially in red meat, are limited (Table 3). Degala et al. [31] reported that ozonated water (pH 6.80, oxygen-reduction potential 562.75 mV, ozone concentration 0.68 mg L^-1^) treatment of chevon samples (20 ± 1 g) for 2, 4, 6, 8, 10, or 12 min resulted in an initial reduction of *E. coli* O157:H7 by 0.19 log_10_ CFU mL^-1^ after 2 min treatment, and by 0.52 log_10_ CFU mL^-1^ after 10 min treatment. However, the possible discoloration of meat at higher ozone concentrations, and ozone’s rapid degradation to oxygen may limit its application in the meat industry.

### 4.3. Electrolyzed Water

Electrolyzed water, now used as a novel nonthermal technology for inactivating microorganisms in the food industry, was first invented in Japan, and since the 1980s, it has been used as a medical product. The strong antimicrobial activity of this environment-friendly technology is due to its pH, oxidation-reduction potential, and chlorine content [32,83]. Electrolyzed water can be produced on site by passing a diluted salt solution through an electrolytic cell that contains anode and cathode electrodes separated by a bipolar membrane. Electrolysis of sodium chloride (NaCl) solution yields Na^+^ and Cl^−^ ions [31,83] and results in acidic electrolyzed water (pH range 2.0–4.0; oxidation-reduction potential > 1000 mV) and alkaline electrolyzed water (pH range 10–11.5; oxidation-reduction potential 800–900 mV).

Electrolyzed water has been used as a decontaminant in a variety of food products such as poultry carcasses, eggshells, salmon fillets, and frozen shrimp. However, studies on its use in red meats are reported for pork and beef, while very little is known about its antimicrobial efficiency in goat meat (Table 3). Arya et al. [76] evaluated the antimicrobial effects of both acidic electrolyzed water and alkaline electrolyzed water on goat meat by spraying samples for different time periods from 2 to 12 min using a household electrolyzed water generator. The authors observed 1.22 and 0.96 log_10_ CFU mL^−1^ reductions in *E. coli* K12 in 12-min acidic and alkaline electrolyzed water treatments, respectively. Although this technology is very effective in decontaminating fresh meat, it can leave chemical residues that can potentially affect meat color and flavor, since it is a chlorine-based technology. Because of its antimicrobial effectiveness, low cost, and ease of operation, electrolyzed water treatment of goat meat may be a viable intervention strategy for small and very small meat processors and retailers [76]. This technology can also be used in conjunction with other nonthermal technologies to further enhance the safety of goat meat and other muscle foods.

**Table 3 animals-11-02943-t003:** Examples of studies on reducing *E. coli* populations in other red meats.

Meat Type	Intervention Method	Microorganism	Reduction	Reference
Beef carcasses	Water wash + aqueous ozone was sprayed on the surface of inoculated sample surface	*E. coli* O157:H7	No effect	[84]
Beef trimmings	1% ozonated water treatment for 7 or 15 min and effect was observed for 0 to 7 days.	*E. coli*	0.64 to 1.05 log CFU g^−1^	[85]
Pork	Meat samples were treated with LcEW * (pH: 6.8; ORP: 700 mV; Chlorine concentration: 100.1 ppm) for 5 min at 23 °C	*E. coli* O157:H7	1.7 log CFU g^−1^	[32]
Beef	Beef samples were treated with AEW ** (pH: 2.3–2.7; ORP ***: 110–1200 mV; Chlorine concentration: 50 ppm) for 3 min at 23 °C	*E. coli* O157:H7	1.6 log CFU g^−1^	[86]
Ground beef	Ground beef samples were treated with 0.5% and 1% citral (essential oil extract of *Melissa officinalis* leaf) for 30 s and stored at 4 °C	*E. coli* cocktail	0.5–1.0 log CFU g^−1^	[87]
Beef patties	Samples were treated with 0.2% of ginger and basilica essential oils	*E. coli*	Significantly lower *E. coli* numbers compared to control	[88]

* LcEW: low-concentrated electrolyzed water; ** AEW: acidic electrolyzed water; *** ORP: oxidation reduction potential.

### 4.4. Ultraviolet Light

Ultraviolet (UV) light is another nonthermal method approved for surface decontamination of food. Both continuous UV light and pulsed UV light are used in the food industry to inactivate pathogens in foods. Ultraviolet light falls in the wavelength range between 100 and 400 nm in the electromagnetic spectrum and is categorized as long wave (UV-A, 320–400 nm), medium wave (UV-B, 280–320 nm), and short wave (UV-C, 200–280 nm). The bactericidal effect of UV light is dependent upon its dose; the best antibacterial effect is between 245 and 285 nm. The ultraviolet dose can be calculated using the formula:D = I × t
where:

D = Ultraviolet light dose, µJ cm^−2^

I = Intensity, µW cm^−2^

t = Exposure time

Ultraviolet light treatment damages the genetic material and cell membrane of bacteria, thus inactivating the organisms. In addition, the release of free radicals during UV light treatment can also damage the cell membrane, enzymes, and nucleic acids of microorganisms. The antimicrobial efficacy of UV light depends on the optical properties of the medium, color, transparency, and the number of soluble and suspended solids [89]. Ultraviolet treatment has been reported to be more effective against gram negative bacteria than gram positive bacteria, yeasts, molds, and viruses, thus UV light may be suitable for inactivating pathogenic *E. coli* in meat.

Pulsed UV light is advantageous compared to other thermal and chemical methods due to its antibacterial efficiency without leaving chemical residues on meat and without causing changes in meat flavor, texture, and nutrient contents [90]. The limitations of pulsed UV light treatment include its ability to inactivate microorganisms present only on the surface of meat. Pulsed UV light induces photochemical, photophysical and/or photothermal damage, resulting in the inactivation or death of bacterial cells [90]. The photophysical inactivation effect of pulsed UV light on bacteria is attributed to the disruption of the cell membrane and components, and the photothermal inactivation effect is due to temporary intracellular heating caused by the absorption of photons [89].

Various foods are processed using UV light at different intensities and treatment times to inactivate pathogenic microorganisms, including *E. coli* O157:H7, as well as to inactivate certain enzymes that promote oxidation and deterioration in foods (Table 3). Degala et al. [30] evaluated the antimicrobial effect of UV light on goat meat by applying intensities of 100 and 200 µW cm^−2,^ with treatment times ranging from 2 to12 min, yielding energy dosages of 0.2–2.4 mJ cm^−2^. The authors found that log reduction of *E. coli* K12 significantly increased when UV light intensity increased from 100 to 200 µW cm^−2^, with a maximum reduction of 1.18 log_10_ CFU mL^−1^ at an intensity of 200 µW cm^−2^ for 12 min. However, increasing the treatment time from 2 min to 12 min did not significantly influence the log reduction. The authors also reported that superficial lipids and proteins in meat with strong UV light-absorbing properties can interfere with the antimicrobial activity of UV light on meat [30]. Bryant et al. [34] assessed the efficiency of pulsed UV-light in inactivating *E. coli* K12 on goat meat and beef surfaces. In this study, the meat samples were placed in the pulsed UV-light sterilization chamber at three distances (4.47, 8.28 and 12.09 cm) from the light source and treated for different time periods from 5 to 60 s. The authors observed a maximum log reduction of 1.66 and 1.74 CFU mL^−1^ on goat meat and beef, respectively, at 4.47 cm distance for 60 s. 

### 4.5. Sonication

Application of ultrasonic waves as an intervention technology to reduce pathogens in foods is generally considered as safe, nontoxic, and environmentally friendly. Ultrasound waves can be classified, based on frequency ranges, into two categories: (i) high power ultrasound (frequency range 20 to 100 kHz) and (ii) low power ultrasound (frequency of 100 kHz or above) [91]. High power ultrasound is regarded suitable for use in food processing and preservation [92]. 

Ultrasonic waves cause intracellular cavitation and thinning that increase the permeability of the cell membrane [93]. Ultrasound waves create negative pressure, resulting in breakage of the cell wall through a series of compression cycles caused by cavitation bubbles that pass through the solution [92]. When these cavitation bubbles collapse, hydroxyl radicals are produced, which recombine to form hydrogen peroxide and molecular hydrogen resulting in the damaging of DNA and thinning of the cell membrane [94]. The antimicrobial efficacy of ultrasound waves, however, depends upon the frequency and amplitude of the ultrasound waves, the temperature and viscosity of the liquid medium, the shape and size of the microorganisms, and the type of cell wall and its physiological state [95]. Ultrasound has been assessed for its antimicrobial efficiency in different types of meat and meat products; however, to the best of our knowledge its use in goat meat has not been reported.

Caraveo et al. [96] observed a 3 log_10_ CFU cm^−2^ reduction in enterobacteria and mesophilic aerobic and psychrophilic bacteria in beef treated with 40 kHz frequency and 11 W cm^−2^ intensity for 60–90 min. In addition to inactivating microorganisms in meat, ultrasonic waves have also been shown to improve texture, margination, water-holding capacity, and cooking-yield of meat [92]. Ultrasound waves have also been used to improve the tenderness and water-holding capacity of beef [97]. The antimicrobial efficacy of ultrasound with a frequency of 40 kHz and intensity of 2.5 W cm^−2^ in distilled water and lactic acid solutions for 3 and 6 min against *S. anatum*, *E. coli*, *Proteus* species and *P. fluorescens* on chicken wing surfaces was investigated [98]. These authors reported a 1.0 log_10_ CFU cm^−2^ reduction in the number of microorganisms on skin surface in water for 3 min, and a greater reduction in bacteria when the time was extended to 6 min. Sonication in the lactic acid aqueous solution for 3 min resulted in more than 1.0 log_10_ CFU cm^−2^ reduction, and after 6 min the reduction exceeded 1.5 log_10_ CFU cm^−2^. 

### 4.6. Low-Voltage Direct Electric Current

The use of low-intensity currents to inactivate bacteria on meat has received some research attention in recent years due to its efficiency, low cost, and ease of application. The mechanisms involved in the inactivation of microorganisms by low-voltage electric current are not completely understood. It is believed that the physical activity of electric currents may inactivate *E. coli*, possibly by disrupting the bacterial cell membrane. It has been reported that the electric current affects the cell membrane orientation and thereby the cell viability [99]. When NaCl is used as a medium to apply low level current, both NaCl and the low-level current can act synergistically in inactivating microorganisms. When electrolyzed, chlorides are converted to chlorine gas, which can play a significant role in inactivation of *E. coli* [100]. 

Different species of microbes can be inactivated using different intensities of direct electric current [100]. Even a low micro amperage can be effective in reducing the number of microorganisms such as *E. coli* and inhibiting their growth [101]. 

Saif et al. [33] applied different intensities (10, 20 and 30 mA/cm^2^) of dc square wave electric signal for different durations (2, 8 and 32 min) to goat meat samples inoculated with *E. coli* O157:H7 and surface-coated with a thin film of 0.15 M NaCl, and found that all three intensities of current were effective in inactivating bacterial cells at a treatment duration of 32 min. The researchers also found that decreasing treatment duration decreased log reduction of *E. coli,* and frequencies of ≥1 kHz and duty cycles of ≥50% accelerated inactivation of the bacteria at a current intensity of 20 mA cm^−2^. Mahapatra et al. [102] used a low-voltage electric inactivation system to apply low voltage dc current to beef samples inoculated with *E. coli* O157:H7 and surface-coated with a thin film of 0.15 M NaCl solution, and observed that increases in current intensity, frequency, duty cycle, and treatment duration increased percent reduction in *E. coli*. However, this study also revealed that the application of a low intensity current can potentially affect sensory properties, such as color and tenderness. Localized heating due to low intensity current can cause discoloration and increase Warner–Bratzler hardness values in meat [102].

### 4.7. Organic Essential Oils

Plants produce secondary metabolites known as essential oils that act as defensive mechanisms against microorganisms [103]. Several essential oils are used in meat preservation due to their antimicrobial and antioxidant activities. Their antimicrobial activity depends on several factors, including chemical structure, pH, temperature, and oxygen level. Although essential oils are generally recognized as safe, achieving high reductions in microbial counts may require higher concentrations or increased treatment time, both of which can have negative effects on the quality characteristics of food [104]. Several authors have studied the antimicrobial effects of essential oils in combination with other intervention technologies in meat and have found significant reductions in bacterial numbers with minimal effects on the quality characteristics of fresh meat. A few researchers have studied the efficacy of essential oils in decontaminating beef (Table 3); however, their uses in other red meats have not been adequately explored.

Degala et al. [30] studied the effects of three different concentrations of lemon grass oil (0.25%, 0.5%, 1.0%) treatment for different time periods from 2 to 12 min on *E. coli* K12 counts on inoculated goat meat and found that the reductions significantly increased with increasing concentrations of lemon grass oil. The authors observed a reduction of 2.16 log_10_ CFU mL^−1^ with 1% lemongrass oil treatment. Other researchers have reported reductions below detection level with a 1.5% lemon grass oil treatment on minced beef and cooked beef patties [105,106]. Although there is clear evidence that increasing treatment intensity using increased lemongrass oil concentration results in better log reductions, Degala et al. [30] found that increasing treatment time from 2 to 12 min did not result in a significant increase in log reductions. Therefore, the researchers recommended lemongrass oil treatment of 1% concentration for 2 min for extending the microbial shelf life of goat meat. Lemongrass (1.56%) has been reported to significantly reduce *L. monocytogenes* in ground beef [107]. 

### 4.8. Hurdle Technologies

Although several technologies are capable of inactivating microorganisms when used individually, they may cause negative effects on the sensory properties of fresh meat, particularly when used in higher treatment strengths. When two or more processes are combined, the microbial shelf life of meat can be increased using the synergistic effects of lower individual treatment intensities and minimum energy input [108]. Combination treatments are likely to have zero, or minimal, negative effects on the sensory properties of meat, such as any texture, color, or flavor deterioration due to lipid oxidation. Furthermore, combination treatments can combat microbial stress adaptation associated with lower (sub-lethal) treatment intensities [109].

Several combination techniques on meat have been studied that involved organic oils, ozonated water, electrolyzed water, ultrasonic waves, and UV light, to name a few. Hurdle treatment of lemongrass oil with cold nitrogen plasma enhanced the antimicrobial activity of lemongrass on pork and minimized the potential negative effects on the sensory properties of the meat [110]. Degala et al. [30] reported that a hurdle treatment comprising of 1% lemongrass oil plus UV light with 200 µW cm^−2^ for 2 min resulted in a synergistic *E. coli* K12 reduction of 6.66 log_10_ CFU mL^−1^ (below detection levels). Ozonated water was used along with electrolyzed water to reduce contamination of cattle hide prior to slaughter [111]. Degala et al. [31] observed an *E. coli* K12 reduction of 0.86 log_10_ CFU mL^−1^ on goat meat with a 12 min combination treatment comprising ozonated water and acidic electrolyzed water. This study also revealed that the combination treatment was significantly more effective than ozonated water treatment alone based on log reductions. 

## 5. Combination of Pre- and Post-Harvest Interventions

Although individual intervention strategies have proven to be effective in reducing pathogens on goat carcasses, a combination of both pre- and post-harvest strategies may be of value in further ensuring food safety in goat carcasses. For example, a dietary manipulation during the weeks prior to slaughter to reduce the number of *E. coli* in the gastrointestinal tracts of goats, followed by postmortem spray washing to decontaminate the skin or carcass, has been demonstrated to be useful. Kannan et al. [16] reported that brown seaweed extract supplementation for two weeks prior to slaughter, combined with chlorinated spray wash during processing, can be used as a viable decontamination strategy in goat processing. Spray washing reduced the total number of bacteria on the skin of goats. Although dietary treatment did not influence skin or dressed carcass bacterial loads, rumen *E. coli* counts were significantly lower in the seaweed extract-supplemented group while rumen pH and VFA concentrations were unaffected. The authors further suggested that the two-step pathogen reduction strategy can be easily implemented in very small meat processing plants with minimum modifications in the existing Hazard Analysis Critical Control Points (HACCP) plans. Ba et al. [23] used organic acid spray treatments, both immediately prior to slaughter (live animal hide) and immediately after slaughter (carcass surface) to decontaminate carcasses. The two-step treatment effectively reduced multiple bacterial species, including *E. coli*, by 2–5 log units, without negatively affecting the meat quality.

## 6. Recommendations

Farmers can adopt certain meat goat management measures that can minimize gastrointestinal *E. coli* populations in addition to improving animal wellbeing and productivity. Goat-handling methods that do not increase stress are of great value in improving animal performance and product quality. Feeding goats on a hay diet a few days prior to shipping them to the processing facility can reduce gut *E. coli* counts. Feeding with high-condensed tannin-containing forages, such as sericea lespedeza in any form (ground, pellet, or hay), may reduce the numbers of certain types of bacteria. Farmers can also consider feed supplements such brown seaweed extract and essential oils derived from herbal plants in meat goat diets, due to their overall positive effect on antioxidant activity, immune function, and product quality characteristics, in addition to reducing gastrointestinal *E. coli* populations.

Processing plant operators must coordinate the overall feed deprivation time meat goats are subjected to prior to processing. Ideally, overnight feed deprivation is recommended, as extended feed deprivation times can increase stress in goats and shorter feed deprivation durations may not result in adequate emptying of gut contents. At the industrial level, spray washing carcasses with organic acids is a proven method to reduce pathogens. Spraying electrolyzed water is another method that could be implemented in processing operations of any scale. At further processing levels, the use of organic essential oils and UV light may be practicable methods depending on the prevailing regulations in different countries. Hurdle technologies can also be adopted for further processed products, although the use of these methods may depend on the type of product, conditions of packaging and storage, and the intended length of shelf life. 

## 7. Conclusions

Goat meat production and processing methods vary from country to country in the developing world, although in most cases goats are raised free range on brush or unestablished pastures and processed following the halal method. Concentrate feeding for meat goats is not common, except in some larger establishments where they are finished on concentrate supplements. In many developing countries, slaughter and meat hygiene practices, as well as regulations and inspections, are still not well established or followed. Small-scale butcher shops without refrigeration facilities, and chevon being sold as bone-in cubed meat are common sights in underdeveloped countries, particularly in the rural areas. However, processing facilities that cater to export goat meat markets likely maintain international hygiene standards and requirements. Reducing initial bacterial contamination in goat carcasses during the slaughter process becomes crucial in controlling the microbial quality of resultant products in regions where there are frequent interruptions in the cold chain.

Any pathogen reduction intervention strategy, pre- or post-harvest, must not compromise animal welfare, live and carcass weights, meat quality characteristics, and other variables of economic importance. For example, an easy-to-apply preharvest management method of hay feeding and shorter feed deprivation period could decrease gut bacterial counts in goats, with no significant effect on their physiological status. Any post-harvest method that increases the temperature of the carcass or cuts can potentially affect the appearance, color, tenderness, flavor, and lipid and pigment oxidation. The nonthermal methods discussed in this paper, irrespective of meat type, could be viable pathogen reduction methods if applied according to recommendations. However, most technologies available are effective for surface decontamination of meat with minimum penetrability. 

The cost-effectiveness and practicality of any pathogen reduction strategy are critical in goat meat production, since the majority of goat processing plants worldwide are small-scale, limited-resource operations. The carcass skin-washing step can be added to the operation with ease with virtually no additional cost to the slaughter facility, but with the advantage of reducing biological hazards in goat carcasses. The additional cost incurred by the producer or processor are offset by the other benefits derived. For example, brown seaweed supplementation may increase goat meat production cost, but it has added benefits, such as better disease resistance in animals, increased antioxidant status, color stability, and shelf life, in addition to reduced fecal shedding of pathogens. Therefore, the application of an intervention strategy may depend on the scale of operation of a production or processing establishment. 

## Figures and Tables

**Figure 1 animals-11-02943-f001:**
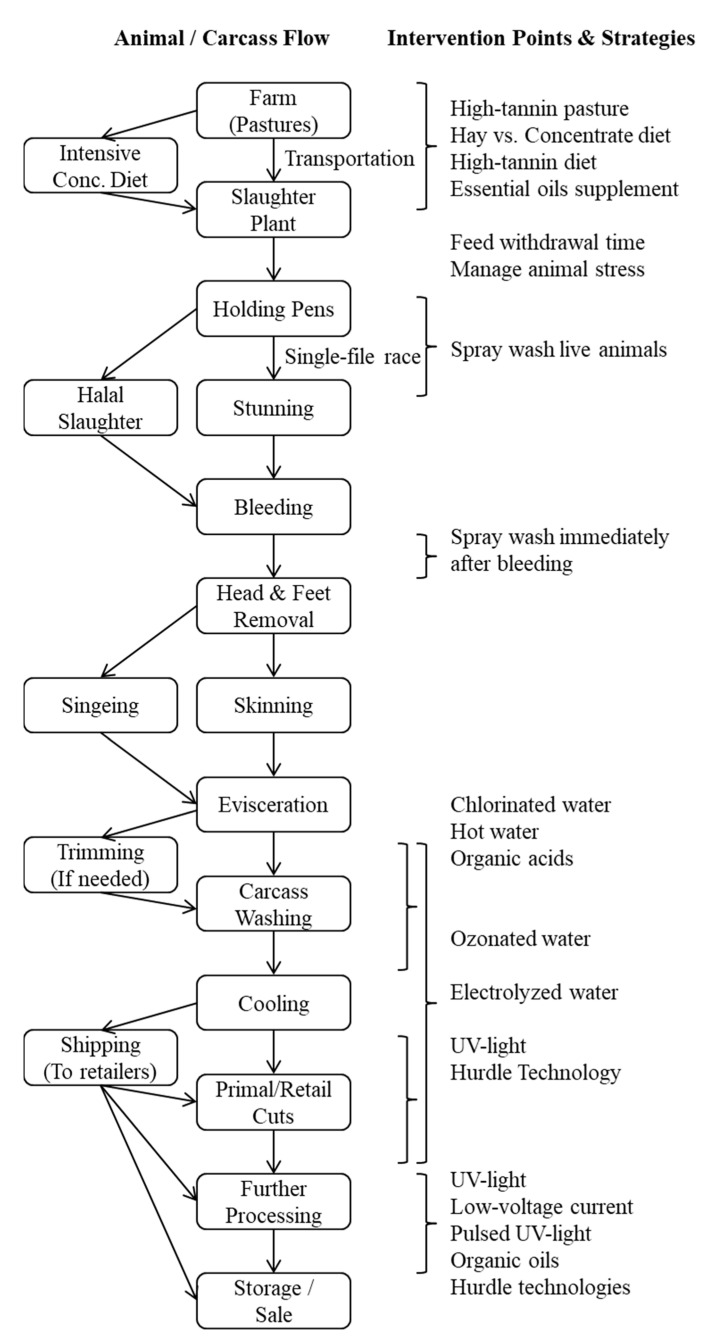
A generic flow diagram of various steps involved in pre- and post-harvest phases of goat meat production to illustrate potential intervention points [12,13,14,15,16,17,18,19,20,21,22,23,24,25,26,27,28,29,30,31,32,33,34].

**Table 1 animals-11-02943-t001:** Reduction in microbial counts resulting from preharvest intervention strategies in goats.

Intervention Stage	Intervention Method	Sample Type	Microorganism	Reduction	Goat Breed	Reference
Holding pens	Spray washing live goats in single file race prior to processing with potable water (0.4–0.8 ppm chlorine; approx. 12 L of water at 15–18 °C per animal) for 1 min.	Skin swab samples	*Generic E. coli* *Aerobic plate count*	No change 0.8 log_10_ CFU cm^−2^	Spanish	[10]
Farm/holding pens	Feed deprivation for 27 h prior to processing (compared to no feed deprivation).	Carcass swab samples	*Generic E. coli* *Aerobic plate count*	0.8 log_10_ CFU cm^−2^ 1.0 log_10_ CFU cm^−2^	Boer × Spanish	[36]
Farm/holding pens	Feed deprivation (in general)	-	*-*	Reduces gut fill and fecal contamination of carcasses	-	[21]
Farm/holding pens	Feed deprivation for 24 h prior to processing (compared to 12 h feed deprivation)	Carcass swab samples	*Generic E. coli* *Aerobic plate count*	No change 0.5 log_10_ CFU cm^−2^	Kiko × Spanish	[11]
Farm	Feeding hay diet for 4 days prior to harvesting (compared to concentrate feeding).	Rectal samples	*Generic E. coli* *Total coliform* *Enterobacteriaceae*	2.4 log_10_ CFU g^−1^ 2.6 log_10_ CFU g^−1^ 3.1 log_10_ CFU g^−1^	Kiko × Spanish	[13]
Farm	Feeding alfalfa hay diet for 90 days prior to processing (compared to concentrate feeding).	Rectal samples	*Generic E. coli*	1.8 log_10_ CFU g^−1^	Boer × Spanish	[37]
Farm	Feeding ground sericea lespedeza (*Lespedeza cuneata*) for 14 weeks.	Fecal samples	*Generic E. coli* *Total plate count*	No change 1.6 log_10_ CFU g^−1^	Kiko × Spanish	[38]
Farm	Feeding brown seaweed (*Ascophyllum nodosum*) supplement for 14 days.	Rumen samples	*Generic E. coli*	1.4 log_10_ CFU g^−1^	Boer × Spanish	[16]
During processing	Spray washing skin for 1 min. using potable water followed by 1 min. with chlorinated water (50 mg L^−1^ hypochlorite) immediately after bleeding.	Skin swab samples	*Aerobic plate count*	1.0 log_10_ CFU cm^−2^	Boer × Spanish	[16]

**Table 2 animals-11-02943-t002:** Reduction in microbial counts resulting from postharvest intervention strategies in goats.

Intervention Stage	Intervention Method	Sample Type	Microorganism	Reduction	Reference
Postharvest	2.5% acetic acid spray for 10 sec. using low-pressure hand sprayer	Carcasses	*Generic E. coli*	1.18 log10 CFU cm^−2^	[75]
Postharvest	Pulsed dc square wave electricity with 10, 20, or 30 mA cm^−2^ current intensities.	Inoculated goat meat samples	*E. coli* O157:H7	8.0 log_10_ CFU mL^−1^	[33]
Further processing	Spraying acidic electrolyzed water for 12 min.	Inoculated boneless goat meat samples	*E. coli* K12	1.2 log_10_ CFU mL^−1^	[76]
Further processing	Spraying alkaline electrolyzed water for 12 min.	Inoculated boneless goat meat samples	*E. coli* K12	0.9 log_10_ CFU mL^−1^	[76]
Further processing	Applying UV-C * for 12 min. at 200 µW cm^−2^	Inoculated boneless goat meat samples	*E. coli* K12	1.2 log_10_ CFU mL^−1^	[30]
Further processing	Spreading 1% lemongrass oil on the surface for 8 min.	Inoculated boneless goat meat samples	*E. coli* K12	2.1 log_10_ CFU mL^−1^	[30]
Further processing	Dipping in ozonated water for 12 min.; pH 6.8, ORP ** (mV) 562.75, ozone concentration 0.68 mg L^−1^.	Inoculated boneless goat meat samples	*E. coli* K12	0.4 log_10_ CFU mL^−1^	[31]
Further processing	Hurdle Technology: Dipping in ozonated water for 6 min. (pH 6.8, ORP (mV) 562.75, ozone concentration 0.68 mg L^−1^) followed by dipping in acidic electrolyzed water for 6 min.	Inoculated boneless goat meat samples	*E. coli* K12	0.86 log_10_ CFU mL^−1^	[31]
Further processing	Hurdle Technology: Spreading 1% lemongrass oil on the surface for 1 min followed by applying UV-C for 1 min. at 200 µW cm^−2^	Inoculated boneless goat meat samples	*E. coli K12*	6.6 log_10_ CFU mL^−1^	[30]

* UV-C: ultraviolet-light; ** ORP: oxidation reduction potential.

## Data Availability

All data referenced in this manuscript are already published.

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
