# Peer review of "Preharvest Management and Postharvest Intervention Strategies to Reduce Escherichia coli Contamination in Goat Meat: A Review"

_animals, 2021, doi:10.3390/ani11102943_

Round 1

Reviewer 1 Report

Dear Authors.

The manuscript titled: Preharvest Management and Postharvest Intervention Strategies to Reduce Escherichia coli Contamination in Goat Meat.

It is an interesting paper, which compiles information on pre-and post-harvest strategies that can reduce or eliminate E.coli colonies in goat meat, as well as ensure animal welfare and thus improve the quality of goat meat. The paper was written incoherent and understandable language.

However, the article needs to be clarified in several places to enrich it and improve its comprehensibility:

-Introduction. Indicate the legal limit of faecal contamination organisms allowed in fresh goat meat.

2.Prevalence of E. coli in Goats. Give an average value of E.coli colonies usually found in both skin and carcass of goats, in order to better understand what levels to start from and what is achieved with each of the intervention strategies applied.

- Does the breed of goat influence the E.coli colony count?  If so, I think it would be interesting to include the breeds of goats and the type of breeding in the justifications and in tables 1 and 2.

3.Pre-Harvest Intervention.

-Why did these authors state that the total e.coli count is more associated with colon pH, please explain the cause in depth?

-Line 201. Why are E.coli organisms very sensitive to pH changes? Explain

3.4. Tannin-Containing Feed Supplements. I think the section on dietary supplementation should be more extensive and not just talk about brown seaweed extract, as an antimicrobial effect has been seen in diets supplemented with other natural extracts or spices and herbs such as rosemary, thyme, or sage.

-Lines 302-303. Insert bibliographic citation

-Lines 316-318. Put bibliographic citation

In addition, I believe that a section on the recommendations regarding the mitigation of Escherichia coli Contamination, both for livestock farmers and at the industrial level would be enriching.

Author Response

Responses to Reviewers’ Comments

Reviewer I

The manuscript titled: Preharvest Management and Postharvest Intervention Strategies to Reduce Escherichia coli Contamination in Goat Meat.

It is an interesting paper, which compiles information on pre-and post-harvest strategies that can reduce or eliminate E.coli colonies in goat meat, as well as ensure animal welfare and thus improve the quality of goat meat. The paper was written in coherent and understandable language.

However, the article needs to be clarified in several places to enrich it and improve its comprehensibility:

  1. Introduction

Indicate the legal limit of faecal contamination organisms allowed in fresh goat meat.

The Agricultural Marketing Service of the US Department of Agriculture has established 500 CFU g-1 as a critical limit for generic E. coli in red meat such as beef.  This point has been included in the Introduction section (Page 2). To our knowledge, a universal legal limit for generic E.coli has not been established for goat carcasses, although different countries have different standards.

  1. Prevalence of E. coliin Goats.

Give an average value of E.coli colonies usually found in both skin and carcass of goats, in order to better understand what levels to start from and what is achieved with each of the intervention strategies applied.

Revised.  Please see “Prevalence of E. coli section (Page 3).

Does the breed of goat influence the E.coli colony count?  If so, I think it would be interesting to include the breeds of goats and the type of breeding in the justifications and in tables 1 and 2.

Controlled studies on the effects on breed on fecal shedding in goats have not been conducted to our knowledge.  This point is included in the Prevalence of E.coli section (Page 3).  However, we have included the breed details in the text and Table 1 as suggested for easy interpretation (Pages 5, 7, and 9).  Breed details were not available for studies cited in Table 2.  Furthermore, we feel breed will not have any effect on the efficacy of postharvest intervention technologies.

  1. Pre-Harvest Intervention.

Why did these authors state that the total E.coli count is more associated with colon pH, please explain the cause in depth?

Revised.  This point has been explained in some detail in the Dietary Regimens and E. coli Populations section (Page 6).

Line 201. Why are E.coli organisms very sensitive to pH changes? Explain

Revised.  Please see Feed Withdrawal and E.coli Populations section (Page 7).

3.4. Tannin-Containing Feed Supplements.

I think the section on dietary supplementation should be more extensive and not just talk about brown seaweed extract, as an antimicrobial effect has been seen in diets supplemented with other natural extracts or spices and herbs such as rosemary, thyme, or sage.

Revised.  This is a great suggestion.  A separate section titled 3.5. Essential Oil-Containing Feed Supplements (Page 8-9).

Lines 302-303. Insert bibliographic citation

Revised. Please see Page 10.

Lines 316-318. Put bibliographic citation

Revised. Please see Page 11.

In addition, I believe that a section on the recommendations regarding the mitigation of Escherichia coli contamination, both for livestock farmers and at the industrial level would be enriching.

Revised. Great suggestion.  A separate section titled 6. Recommendations (Page 17) has been added.

Reviewer 2 Report

This paper is good presented and results are important for environmental studies.
There are some typological and grammatical errors. It should be corrected.
1. Vendor information for instruments and chemicals must be given consistently and completely: (company and country name).
2. Spell of references must be checked.
3. RSD values should be added to abstract.
4. LOD and LOQ values msut be defined.
5. Some important figures must be obeyed.

Reviewer 3 Report

The article includes a review of 111 references (from 1991 to 2021) of possible interventions to reduce the contamination of "E. coli" in goat meat.

In my opinion, the paper is well structured, instructive, and exhaustive enough that it may be useful for further studies.

In my opinion, it should be expressly indicated in the title and in the abstract that the article is a review.

References 45 and 89 must put the year in bold

Round 2

Reviewer 2 Report

I recommended that this paper should be accepted for publication.
